# Effects of Aerobic and Resistance Exercise on Myokines in High Fat Diet-Induced Middle-Aged Obese Rats

**DOI:** 10.3390/ijerph17082685

**Published:** 2020-04-14

**Authors:** Nayoung Ahn, Kijin Kim

**Affiliations:** Department of Physical Education, Keimyung University, Daegu 42601, Korea; nyahn13@kmu.ac.kr

**Keywords:** exercise, obesity, myokine, aging, skeletal muscle, rat

## Abstract

The objective of this study was to analyze the effects of aerobic and resistance exercise on myokines expression in the skeletal muscle of middle-aged rats with high fat diet-induced obesity, to investigate the feasibility of using exercise training to reduce inflammation. Male 50-week-old Sprague Dawley rats were divided into normal diet, normal diet + exercise, high fat diet, and high fat diet + exercise groups. After six weeks on a high fat diet to induce obesity, a 12-week exercise program was implemented, which combined aerobic exercise (treadmill running) and resistance exercise (ladder climbing) three times a week for 75 min per session. We analyzed the protein levels of interleukins (IL) 6, 7, and 8, C-X-C motif chemokine receptor 2, and vascular endothelial growth factor in skeletal muscles by western blotting. Body weight decreased significantly during the 12-week exercise program in the exercise groups compared to the non-exercise groups (*p* < 0.05). The levels of all myokines analyzed were significantly lower in the skeletal muscle of the high fat diet group compared to the normal diet group (*p* < 0.05). After completing the 12-week exercise program, IL-7, IL-8, C-X-C motif chemokine receptor 2, and vascular endothelial growth factor expressions were significantly higher in the high fat diet + exercise group compared to the high fat diet group (*p* < 0.05). However, while IL-6 expression was significantly lower in the high fat diet and high fat diet + exercise groups compared to the normal diet group (*p* < 0.05), it was not significantly affected by exercise. In conclusion, high fat diet-induced obesity resulted in decreased myokines in the skeletal muscles, but combined exercise training of aerobic and resistance exercise increased myokines secretion in the skeletal muscle of obese rats, and is thought to help reduce inflammation.

## 1. Introduction

Aging-induced sarcopenia can cause immunosenescence and significantly impact the onset of metabolic diseases, including cardiovascular disease. In particular, the increased secretion of proinflammatory adipocytokines caused by sarcopenic obesity promotes muscle catabolism [1]. Skeletal muscle secretes various proteins, including growth factors, cytokines, and metallopeptidases, and muscle contraction during exercise training induces myogenesis by increasing the expression of myokines, muscle-derived molecules that mediate the “polypill” effect of exercise [2]. The secretion of myokines, which include angiopoietin-like 4, brain derived neurotrophic factor, fibroblast growth factor (FGF) 21, follistatin-like 1, interleukin (IL)-6, IL-7, IL-15, irisin, LIF interleukin 6 family cytokine (LIF), myonectin, myostatin, and vascular endothelial growth factor (VEGF), varies upon muscle contraction [3].

Along with increased myokine expression, muscle contraction during exercise triggers neovascularization through potent angiogenic factors, such as IL-8 and VEGF [4]. Myokine expression levels are a key indicator of immune function, and IL-8 is a major myokine [5] that facilitates angiogenesis through C-X-C motif chemokine receptor (CXCR) 2 signaling [6]. IL-8 is a chemokine that stimulates the recruitment of neutrophils, and the plasma IL-8 concentration increases during strenuous exercise, such as sprinting and eccentric muscle contraction exercises [7]. IL-8 secretion from muscles stimulates neovascularization, and in humans, it imparts its chemokine effects through CXCR1 signaling in microvascular endothelial cells and neovascularization through CXCR2 [8]. CXCR2 is highly expressed in muscle fibers and vascular endothelial cells and displays increased expression in muscle biopsy samples after concentric exercise, suggesting a key role in neovascularization [5]. IL-7 strengthens immune function in two ways: by stimulating T cell production in the thymus, and by enhancing T cell development. In cancer and metabolic diseases, increased IL-7 levels stimulate T cell production in the presence of reduced CD4^+^ T cells due to weakening of the autoimmune system [9]. IL-7 is produced in the stromal cells of lymphoid organs, mainly in the thymus in rats [10], and in the spleen in humans [11]. IL-7 production likely occurs at a constant rate, as it has been suggested that circulating IL-7 concentration in the body is regulated by modulating its binding to IL-7 receptors rather than its production [12]. However, IL-7 secretion in the livers of mice is associated with an acute response that facilitates increases in naive and memory CD4^+^ and CD8^+^ T cells [13], in particular, IL-7 has been reported as a novel marker associated with muscle fiber hypertrophy after resistance exercise [14]. Therefore it is expected that it will have a positive effect even after combined exercise training of resistance and aerobic exercise in high fat diet (HFD) induced obese subjects. It is considered that there is a need to confirm these effects more accurately.

IL-6 has been suggested as one of the representative myokine of muscle in relation to the positive effects of chronic disease prevention, including anti-inflammatory effect and efficient insulin action by exercise training. However, IL-6 is reported as a positive factor in relation to muscle hypertrophy [15] and metabolic function activation [16], whereas the result does not show any change in blood IL-6 concentration [17] and an insignificant expression level of mRNA of IL-6 in muscle [14] after exercise training. Therefore, it is considered that detailed confirmation of IL-6 expression in muscle is required after exercise training in high-fat diet induced obese subjects.

While exercise has important pro- and anti-inflammatory effects [2], its effects on the levels of various myokines associated with the prevention of obesity and aging remain unclear. Accordingly, we examined the effects of a combined 12-week exercise training program consisting of aerobic and resistance exercise on aging rats with high fat diet-induced obesity on skeletal muscle myokines expression. The results suggest that exercise training is a promising treatment to reduce inflammation.

## 2. Methods

### 2.1. Subjects

Experiments were performed using forty 50-week-old male Sprague Dawley rats. After acclimating the rats to their environment for one week, obesity was induced with six weeks of a high fat diet, followed by the implementation of a 12-week aerobic and resistance exercise program (Figure 1). The rats were given ad libitum access to food and water. This study was approved by the Animal Experiment Ethics Committee of Daegutechnopark Biohealth Center (BHCC-IACUC-2018-01). Subjects were divided into normal diet (ND), normal diet + exercise (ND+Ex), high fat diet (HFD), and high fat diet + exercise (HFD+Ex) groups (*n* = 10 each). The high fat diet consisted of 30% carbohydrates, 50% fat, and 20% protein based on total calories. The normal diet consisted of 64.5%, 11.8%, and 23.5% carbohydrates, fats, and proteins, respectively, and contained 3.2 kcal/g.

### 2.2. Exercise Program

The exercise program, focused on prevention of obesity and aging, comprised combined exercise consisting of aerobic exercise using a treadmill running [18] and resistance exercise with ladder climbing [19] for the objectives of energy consumption, improvement in cardiovascular function, and increase in muscle strength. The 12-week exercise program consisted of a total of 75 min per daily session and three sessions per week. Exercise was carried out together in one day, resistance exercise was conducted first, followed by aerobic exercise. Rats were anesthetized with Zoletil 50 (10mg/kg body weight) and 2% Rompun (0.04 mL/kg) and anterior tibialis muscles were dissected out 48 h after exercise training for protein analysis.

### 2.3. Protein Isolation and Immunoblotting

Total proteins were extracted using RIPA lysis buffer (Cat. #MB-030-0050, Rockland) containing protease inhibitor cocktail (Cat. #PPI1015, Quartett), and 10 μg of protein was resolved by sodium dodecyl sulfate-polyacrylamide gel electrophoresis and transferred to nitrocellulose membranes using the Trans-Blot Turbo Transfer System (Bio-Rad, Hercules, CA, USA). The membranes were blocked with Tris-buffered saline containing 5% skim milk (Bio-Rad) and 0.2% Tween 20 (Bio-Rad). The following primary antibodies were used: IL-6 (1:500, Santa Cruz Biotechnology Inc., sc-1265), IL-7 (1:500, Biorbyt, orb214102), IL-8 (1:500, Biorbyt, orb229133), CXCR2 (1:500, Biorbyt, orb229230), VEGF (1:500, Santa Cruz Biotechnology Inc., sc-152) and glyceraldehyde 3-phosphate dehydrogenase (GAPDH, 1:3000, Cell Signaling Technology, #2118). After reaction with horseradish peroxidase-conjugated secondary antibodies (Santa Cruz Biotechnology Inc.), protein bands were visualized using Clarity Western ECL Substrate (Bio-Rad) following the manufacturer’s procedure. Band densities were determined using a ChemiDoc XRS+ System (Bio-Rad) and were normalized to GAPDH, which served as a loading control.

### 2.4. Statistical Analysis

Data are presented as the mean ± standard error (SE). The statistically significant difference test for weight gain was performed by two-way analysis of variance (ANOVA) with repeated measures for groups and time. In the case of a significant interaction effect between groups and time, independent t-test between groups and contrast for baselines by group were conducted to determine significance. One-way ANOVA was used to test for significant differences in myokine levels between groups, and Bonferroni post hoc tests were conducted to determine significance, which was set at *p* < 0.05. Statistical analyses were performed with SPSS 22.0 (SPSS Inc., Chicago, IL, USA).

## 3. Results

### 3.1. Body Weight

In the results of the two-way variance analysis of changes in body weight, significant differences were found in the groups (F = 13.343, *p* < 0.001) and time (F = 249.165, *p* < 0.001) of the main factors, and there was a significant (F = 22.254, *p* < 0.001) interaction effect between the groups and time. Obesity was induced for six weeks in the HFD and HFD+Ex groups, which displayed significant increases in body weight compared to the ND groups starting in the second week (Figure 2). During the 12-week exercise program, the ND+Ex and HFD+Ex groups displayed significant decreases in body weight compared to the ND and HFD groups, respectively. By the eighth week of the exercise program, the mean body weight in the HFD+Ex group was no longer significantly different to that of the ND group. The ND + EX group showed significantly lower body weight than the ND group from first week (7th week in total) after the start of exercise treatment, and The HFD + EX group showed no significant difference from the ND group from the 8th week of exercise treatment (14th week in total). Through these results, it was possible to confirm the exercise effect of clear weight loss.

### 3.2. Myokines Expression in Skeletal Muscle

We next examined the levels of various myokines in the different groups by western blot analysis of skeletal muscle lysates (Figure 3A). IL-7 expression in skeletal muscles was significantly lower in the HFD group compared to the ND group (Figure 3B). After 12 weeks of exercise training, IL-7 expression was significantly higher in the HFD+Ex group compared to the ND and HFD groups and significantly higher in the ND and ND+Ex groups compared to the HFD group. However, there were no significant differences in IL-7 expression between the ND and ND+Ex groups after 12 weeks of exercise training. IL-8 expression in skeletal muscle was significantly lower in the HFD group than in the ND group and significantly higher in the HFD+Ex group than in the ND and HFD groups, revealing a similar pattern to IL-7 (Figure 3C). Moreover, the ND+Ex group had significantly higher IL-8 expression than the ND group, while the ND and ND+Ex groups had significantly higher expression than the HFD group. IL-6 expression was significantly lower in the HFD and HFD+Ex groups than in the ND group (Figure 3D). CXCR2 expression was significantly lower in the HFD group than in the ND group, and after 12 weeks of exercise training, the HFD+Ex group displayed significantly higher CXCR2 expression than the HFD group (Figure 3E). VEGF expression was significantly lower in the ND+Ex and HFD groups compared to the ND group, and after 12 weeks of exercise training, the HFD+Ex group showed significantly higher VEGF expression than the HFD group (Figure 3F).

## 4. Discussion

The anti-inflammatory effects of exercise training are widely recognized to be based on appreciable reductions in pro-inflammatory cytokine concentrations. The prevailing view is that these effects could be beneficial for the prevention and treatment of chronic diseases associated with low-grade systemic inflammation, especially obesity, insulin resistance, cardiovascular diseases, atherosclerosis, and neurodegenerative disorders [20,21]. Moreover, in human skeletal muscles, there are reports of increased production and secretion of various cytokines after exercise training, including IL-6, IL-8, IL-10, IL-15, CC-chemokine ligand 2 (CCL2), and IL-1 receptor, as well as increased VEGF expression [3]. This study demonstrates that an exercise program combining aerobic and resistance exercise activates the expression of several myokines, including IL-7, IL-8, CXCR2, and VEGF, in the skeletal muscle of aging obese rats.

IL-7, a myokine secreted in response to skeletal muscle contraction during exercise training, is an anti-inflammatory cytokine involved in regulating muscle hypertrophy, with particularly important effects on myogenesis and migration [14]. In a previous study, the expression of IL-7 mRNA in skeletal muscle was significantly enhanced two weeks after high-intensity exercise training, and remained high 11 weeks later [14]. Moreover, plasma IL-7 concentrations in elite female soccer players significantly increased immediately after a 90-min soccer match, but not following a second soccer match after 72 h of recovery time [22]. In this study, IL-7 expression in the skeletal muscle of the HFD group was significantly lower than in the ND group, but increased significantly in the HFD+Ex group after 12 weeks of exercise training. These results indicate that obesity decreases IL-7 expression, and that exercise training increases it. However, the effects of exercise on IL-7 levels were not observed in rats fed a normal diet. When looking at the results of IL-7 shown in this study, it is expected that the expression pattern may be lowered in the obese state, and the expression level may be activated through exercise to prevent the reduction of muscle mass. However, it is considered that this action does not show a significant difference in the normal diet. As IL-7 affects myogenesis, which requires the expression of specific genes and the replication of satellite cells [15,23], increased expression in the HFD+Ex group supports a potential role for IL-7 in preventing muscle fiber atrophy. However, it is unclear if this effect is due to direct effects on IL-7 gene transcription or the inhibition of terminal differentiation.

IL-6 is a typical myokine known to regulate multiple physiological functions after exercise training, and activates signal transducer and activator of transcription 3 signaling in human satellite cells after muscle-lengthening contraction [16]. Moreover, IL-6 has important effects on hypertrophic muscle growth and myogenesis in mice [15]. In this study, IL-6 protein expression was appreciably lower in the skeletal muscles of obese rats compared to those fed a normal diet; however, although IL-6 increased slightly in obese rats after exercise training, the effect was not significant. Consistent with this, previous studies reported no significant changes in IL-6 mRNA levels after 11 weeks of exercise training [14] nor on plasma IL-6 levels after 12 weeks of endurance training [17]. Taken together, there appears to be limited evidence to suggest that IL-6 levels in skeletal muscles are influenced by exercise [7]. However, the secretion of IL-6 from skeletal muscle after exercise has important effects on the regulation of endocrine function, playing roles in insulin-stimulated glucose disposal and increased glucose oxidation [24], and stimulating lipolysis and fat oxidation [25,26]. At the molecular level, such effects are mediated by the IL-6-dependent activation of AMP-activated protein kinase [27], insulin receptor substrate-1 [28], and phosphoinositide 3-kinase [29]. Moreover, IL-6 also facilitates the alternative activation of macrophages that have been restrained in tissues by obesity-induced insulin resistance and inflammation [30]. In other words, IL-6 secretion after exercise training can have beneficial effects, including blood sugar regulation, lipolysis, inhibition of tumor growth, and maintenance of muscle mass. To definitively identify the effects of exercise training on muscle IL-6 protein expression, additional studies that consider the detailed contents of the exercise program and the specific characteristics of the subjects will be necessary. In addition to the results of various previous studies, it is considered that additional studies are needed in this study to present clearer results as the expression of IL-6 is not significant after exercise training.

IL-8 activates capillary tissues and the proliferation of vascular endothelial cells [31]. In particular, it is a key angiogenesis-facilitating myokine [5] that signals through CXCR2 [6], which have been suggested to play an important role in the formation of skeletal muscle blood vessels. Exercise has important effects on cardiovascular functions, including increasing cardiac output and blood volume, improving oxygen supply and use, and increasing skeletal muscle capillaries, as muscle contraction results in the secretion of factors associated with various physiological functions [32]. It has been widely reported that the expression level of VEGF, an important angiogenic factor, is activated by acute exercise [33]. Obesity leads to decreased skeletal muscle capillary density due to decreased angiogenesis markers; however, exercise training normalizes VEGF signaling and is therefore suggested to be an important therapeutic strategy for vascular disorders [34]. Ballard [35] proposed that the increased neurogenesis and skeletal muscle VEGF induced by exercise may underlie its beneficial effects on neural function.

In this study, the skeletal muscle expression of IL-8, CXCR2, and VEGF was lower in obese rats compared to non-obese rats, consistent with the notion that obesity negatively impacts angiogenesis. Moreover, exercised obese rats displayed significant increases compared to non-exercised obese rats, confirming the increased expression of IL-8, CXCR2, and VEGF in muscles as a result of exercise training. However, Brenner et al. [36] did not detect differences in growth factor levels (VEGF & FGF2) with increasing doses of exercise. Plasma VEGF concentrations do not show distinct changes at the systemic level in response to exercise, displaying various differences according to the intensity and nature of exercise, usually with localized effects [37]. In addition, Amir et al. [38] reported conflicting results in human myotubes, where IL-8 secretion impaired capillary growth. When knee extensions or rowing was performed for 30 min, cytokine levels in the muscular interstitial fluid increased, but this may have been unrelated to increased transcription [39,40]. Moreover, increases in the transcripts and proteins of various secreted factors, including cellular communication network factors 1 and 2, IL-8, IL-15, LIF, and VEGF, do not always produce proportional increases in systemic concentrations [37,41,42,43,44]. Generally, systemic cytokine responses appear more prominently after exercise that causes greater muscle damage, such as downhill running, eccentric exercise, and resistance training [45]. Moreover, robust elevation in the transcripts of various cytokines and chemokines appear in skeletal muscles after long-term or high-intensity exercise [46,47,48], but at most, only slight elevations are observed after moderate-intensity exercise [41,49]. In this study, while the expression patterns of IL-8, CXCR2, and VEGF decreased in obesity, it was found to be markedly activated after exercise training. Considering the previous results showing various differences in the degree of expression according to changes in exercise types including exercise intensity, more detailed studies related to this should be continued. Moreover, the exact cause of the decrease in the ND + EX group compared to the ND group in relation to VEGF expression could not be confirmed and should be left as a research problem for the next study.

Systemic decreases in pro-inflammatory cytokines after exercise are associated with decreased visceral fat mass, increased production and secretion of IL-10 and IL-1 receptor antagonists, which are anti-inflammatory cytokines [50], downregulation of toll-like receptor signaling [51,52], and activation of immune suppressive T cells [53]. Myokines, including IL-7, contribute to these effects [54,55], but the anti-inflammatory effects of exercise are likely mediated through a network of closely interacting factors, making it difficult to isolate the contributions of specific myokines.

Our results indicate that an exercise program combining aerobic and resistance exercise enhances the secretion of anti-inflammatory myokines in obese middle-aged rats. However, myokines secretion patterns seem to depend greatly on the type, intensity, and duration of exercise, and the effects of exercise-induced myokines secretion on inflammation associated with metabolic diseases, as well as in normal immune function, remain unclear. Future studies examining specific exercise modalities will help definitively identify the anti-inflammatory effects of exercise training.

## 5. Conclusions

In conclusion, high fat diet-induced obesity suppressed the protein expression of IL-7, IL-8, IL-6, CXCR2, and VEGF in skeletal muscle. However, combined exercise training of aerobic and resistance exercise increased myokines secretion in skeletal muscle of obese rats, and is thought to help reduce inflammation.

## Figures and Tables

**Figure 1 ijerph-17-02685-f001:**
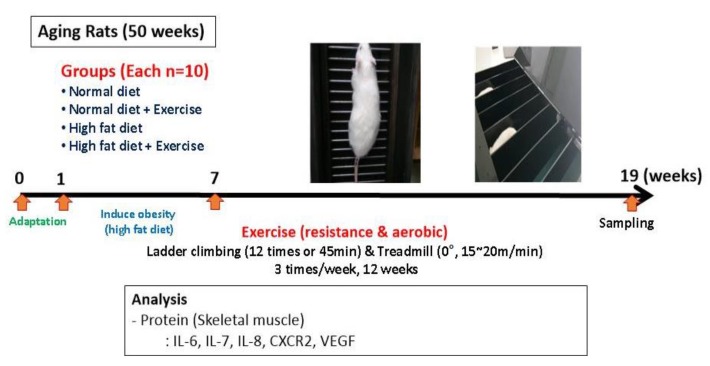
Schematic of experimental design.

**Figure 2 ijerph-17-02685-f002:**
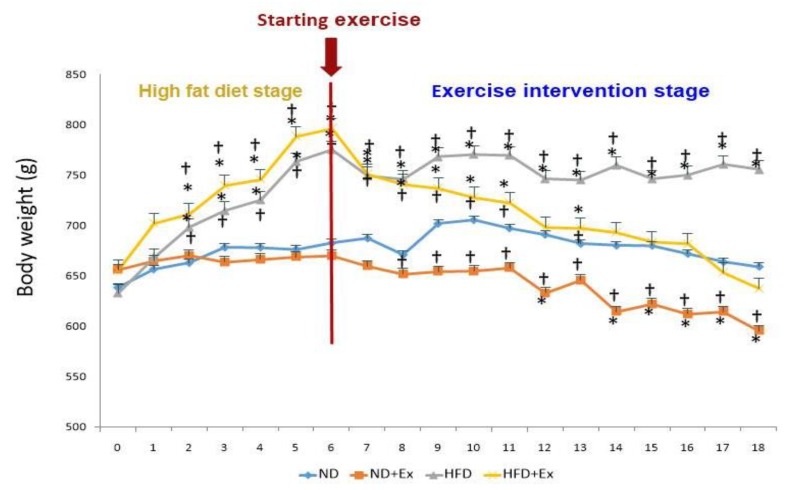
Changes in body weight during high fat diet-induced obesity induction and exercise intervention. *, *p* < 0.05 vs. baseline by paired *t*-test; †, *p* < 0.05 vs. the ND group by independent *t*-test. ND, normal diet; ND+Ex, normal diet + exercise; HFD, high fat diet; HFD+Ex, high fat diet + exercise.

**Figure 3 ijerph-17-02685-f003:**
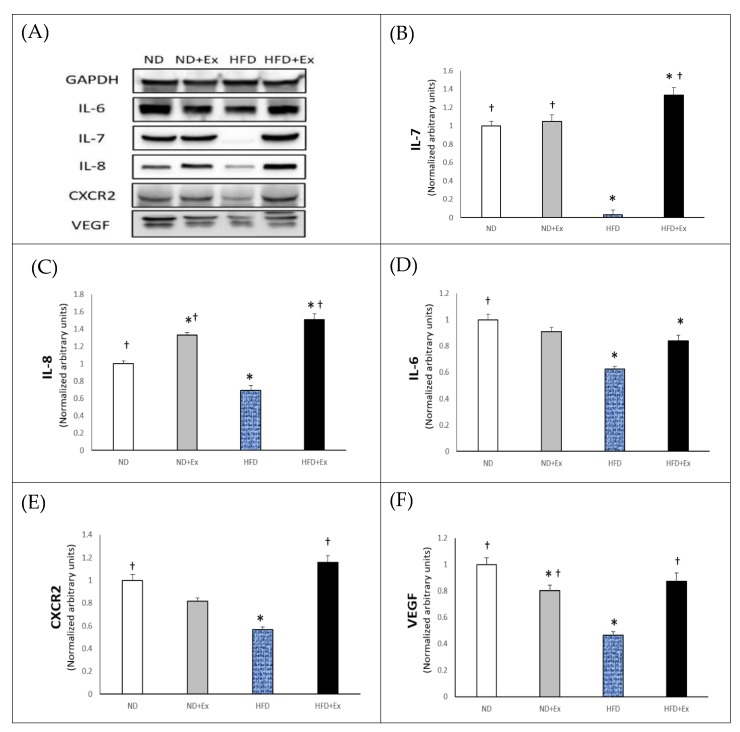
Comparison of myokines protein expression in the normal diet, normal diet + exercise, high fat diet, and high fat diet + exercise groups. (**A**) Myokines levels in skeletal muscle lysates were analyzed by western blot. (**B**–**F**) IL-7, IL-8, IL-6, CXCR2, and VEGF levels. Results represent the mean ± SE. *, *p* < 0.05 vs. the ND group by one-way ANOVA; †, *p* < 0.05 vs. the HFD group by one-way ANOVA. ND, normal diet; ND+Ex, normal diet + exercise; HFD, high fat diet; HFD+Ex, high fat diet + exercise.

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
