# Peer review of "Effects of Aerobic and Resistance Exercise on Myokines in High Fat Diet-Induced Middle-Aged Obese Rats"

_ijerph, 2020, doi:10.3390/ijerph17082685_

Round 1

Reviewer 1 Report

This study evaluated the effects of combined exercise on myokine expression in skeletal muscles (SkM) in middle-aged obese rats fed with high fat diet. Authors presented the significant effects of their exercise program on several SkM myokines, including IL-6, -7, -8, CXCR-2, and VEGF, and concluded exercise reduced inflammation and enhanced immune function during obesity. However, there are several major concerns to conclude.

First, authors did not present evidences that their exercise program improved immune function. Authors argued that the significant changes of several myokines implicated the effects of exercise on immune function in obesity. However, the results is not a direct evidence.

Secondly, in discussion, almost paragraphs are not related to their results, but redudant description for general function of myokines that has been published. How can the results support authors' argue?

Thirdly, the description about exercise protocols in the text is not matched with illustration of Figure 1. In addition, why they used paired t-test even they used two-way ANOVA? Authors should describe the results of repeated measure of ANOVA, not necessary to present the result of paired t-test, in other words, authors have to present interaction of timexgroups.

Finally, in Figure 3, the results of graph for IL-8 was not matched with Western blot data.

This manuscript should be vigorously improved to argue the authors' conclusion.

Author Response

Point 1: First, authors did not present evidences that their exercise program improved immune function. Authors argued that the significant changes of several myokines implicated the effects of exercise on immune function in obesity. However, the results is not a direct evidence.

Response 1: It is considered to be an appropriate point, and since such evidence was not presented, such a description could not be made in the discussion. And we revised as follow in abstracts and conclusion.

In Abstracts :

(Line 8-9) The objective of this study was to analyze the effects of aerobic and resistance exercise on myokines expression in the skeletal muscle of middle-aged rats with high fat diet-induced obesity, to investigate the feasibility of using exercise training to reduce inflammation and enhance immune function.

(Line 24) In conclusion, obesity resulted in decreased myokines in the skeletal muscles of aging rats, but exercise training increased myokines secretion, suggesting its promise in reducing inflammation and enhancing immune function during obesity.

 In Introduction :

(Line 65) The results suggest that exercise training is a promising treatment to reduce inflammation and enhance immune function.

 In 5. Conclusion :

(Line 252) In conclusion, high fat diet-induced obesity suppressed the expression of IL-7, IL-8, IL-6, CXCR2, and VEGF in the skeletal muscles of aging obese rats; however, a 12-week program combining aerobic and resistance exercise increased the secretion of IL-7, IL-8, CRCX2, and VEGF, indicating that exercise is a useful means of reducing inflammation and enhancing immune function.

Point 2: Secondly, in discussion, almost paragraphs are not related to their results, but redudant description for general function of myokines that has been published. How can the results support authors' argue?

Response 2-1 : Considered as appropriate intellectual property, the following description was added to the results of this study in the discussion.

(Line 167-170) ‘When looking at the results of IL-7 shown in this study, it is expected that the expression pattern may be lowered in the obese state, and the expression level may be activated through exercise to prevent the reduction of muscle mass. However, it is considered that this action does not show a significant difference in the normal diet.’

(Line 208-210) ‘In addition to the results of various previous studies, it is considered that additional studies are needed in this study to present clearer results as the expression of IL-6 is not significant after exercise training.’

Response 2-2 : In addition, the following sentence was revised as follows.

(Line 236-238) Therefore, to clearly determine the immune-activating and anti-inflammatory effects of exercise programs of various intensities, continued measurement of cytokine expression levels in skeletal muscles may be necessary.

  • In this study, while the expression patterns of IL-8, CXCR2, and VEGF decreased in obesity, it was found to be markedly activated after exercise training. Considering the previous results showing various differences in the degree of expression according to changes in exercise types including exercise intensity, more detailed studies related to this should be continued.

Point 3: Thirdly, the description about exercise protocols in the text is not matched with illustration of Figure 1. In addition, why they used paired t-test even they used two-way ANOVA? Authors should describe the results of repeated measure of ANOVA, not necessary to present the result of paired t-test, in other words, authors have to present interaction of timexgroups.

Response 3-1 : We think it is a proper point. In the figure, the start time of exercise treatment is indicated. In addition, the statistical processing was reprocessed to add interaction of timexgroups to the result description, and it was revised as a contrast test for the baseline rather than a paired t-test as follow.

Increases in body weight were analyzed by two-way analysis of variance (ANOVA) with repeated measures, and independent and paired t-tests were conducted to determine significance.

  • The statistically significant difference test for weight gain was performed by two-way analysis of variance (ANOVA) with repeated measures for groups and time. In the case of significant interaction effect between groups and time, independent t-test between groups and contrast for baselines by group were conducted to determine significance.

Response 3-2 : In addition, the following was added in the results description.

‘In the results of the two-way variance analysis of changes in body weight, significant differences were found in the groups (F=13.343, p<0.001) and time (F=249.165, p<0.001) of the main factors, and there was a significant (F=22.254, p<0.001) interaction effect between the groups and time.’

Point 4: Finally, in Figure 3, the results of graph for IL-8 was not matched with Western blot data.

Response 4 : The error in Figure 3 has been confirmed and revised.

Point 5: This manuscript should be vigorously improved to argue the authors' conclusion.

Response 5 : The conclusion section was revised as follows.

(Line 252) In conclusion, high fat diet-induced obesity suppressed the expression of IL-7, IL-8, IL-6, CXCR2, and VEGF in the skeletal muscles of aging obese rats; however, a 12-week program combining aerobic and resistance exercise increased the secretion of IL-7, IL-8, CRCX2, and VEGF, indicating that exercise is a useful means of reducing inflammation and enhancing immune function.

Reviewer 2 Report

Ahn and Kim report the effects of aerobic and resistance exercise on selected myokines of middle aged rats.

Overall, the presented work is interesting and can add further knowledge on the effects of exercise to middle-aged rats.However, based on the data provided the authors should, perhaps, consider to publish this as a short communication rather than a full paper.

Major points:

In the introduction (lines 50-59), the authors describe the properties of IL-7, but do not explain, or introduce hypotheses, on how its production can be correlated with exercise (mention citation 18?).

In the introduction, the authors do not mention IL-6 that is analysed in the results section, and thoroughly discussed in the discussion. Il-6 should me mentioned and described equally to IL-7 and 8 to provide a more complete set of information.

The Exercise program is not clear: are the aerobic and resistance exercises performed the same day and in what order?

In section 2.4 is not clear what is a repetition and what a set (lines 103-107).

Are the body weights in HFD and HFD+EX, ND and ND+EX significantly different? The authors write this in the text (lines) but do not show or explain properly in figure 2.

The graph in figure 2 should also report the standard error.

The western blot in figure 3A seems not in agreement with the quantitation of IL-8 in graph 3C. The immunoband in ND+EX seems less intense than HFD (and GAPDH is similar for all the 4 samples).

It would be interesting and helpful to calculate the ratio of IL-8/CXCR2 to understand whether the receptor expression paralleled the increase of IL-8.

The discussion on IL-6 seems to me rather long. The main conclusion is that IL-6 does not change and is in agreement with previous data reported in the literature.

How can the decrease of the VEGF in ND+EX group be explained?

Minor points:

Figure 1: climb down is not correct

Line 97: slope or inclination is more appropriated than incline

Line 112: Clipped left ventricle? I think this sentence is not correct

The references present some typos errors.

Author Response

Point 1: Overall, the presented work is interesting and can add further knowledge on the effects of exercise to middle-aged rats. However, based on the data provided the authors should, perhaps, consider to publish this as a short communication rather than a full paper.

Response 1: I agree with your opinion, but I would like to submit the original paper if possible. But I can follow that if you strongly want.

Major point:

Point 2: In the introduction (lines 50-59), the authors describe the properties of IL-7, but do not explain, or introduce hypotheses, on how its production can be correlated with exercise (mention citation 18?).

Response 2: We revised as follow.

(Line 56-59) However, IL-7 secretion in the livers of mice is associated with an acute response that facilitates increases in naive and memory CD4+ and CD8+ T cells [13], and thus, chronic exercise may have different effects on IL-7 expression than acute exercise [7].

  • However, IL-7 secretion in the livers of mice is associated with an acute response that facilitates increases in naive and memory CD4+ and CD8+ T cells [13], in particular, IL-7 has been reported as a novel marker associated with muscle fiber hypertrophy after resistance exercise [14]. Therefore it is expected that it will have a positive effect even after combined exercise training of resistance and aerobic exercise in HFD-diet induced obese subjects. It is considered that there is a need to confirm these effects more accurately.

Point 3: In the introduction, the authors do not mention IL-6 that is analysed in the results section, and thoroughly discussed in the discussion. Il-6 should me mentioned and described equally to IL-7 and 8 to provide a more complete set of information.

Response 3 : The following sentence was added in the introduction.

‘IL-6 has been suggested as one of the representative myokine of muscle in relation to the positive effects of chronic disease prevention, including anti-inflammatory effect and efficient insulin action by exercise training.

However, IL-6 is reported as a positive factor in relation to muscle hypertrophy [15] and metabolic function activation [16], whereas the result does not show any change in blood IL-6 concentration [17] and the no significant expression level of mRNA of IL-6 in muscle [14] after exercise training. Therefor it is considered that detailed confirmation of IL-6 expression in muscle is required after exercise training in high-fat diet induced obese subjects.’

Point 4: The Exercise program is not clear: are the aerobic and resistance exercises performed the same day and in what order?

Response 4: It was added to exercise program as follows.

‘Exercise was carried out together in one day, resistance exercise was conducted first, followed by aerobic exercise.’

Point 5: In section 2.4 is not clear what is a repetition and what a set (lines 103-107).

Response 5: Here, the repetition period means a comparison between before and after. In addition, the statistical process was revised as follows.

Increases in body weight were analyzed by two-way analysis of variance (ANOVA) with repeated measures, and independent and paired t-tests were conducted to determine significance.

  • The statistically significant difference test for weight gain was performed by two-way analysis of variance (ANOVA) with repeated measures for groups and time. In the case of significant interaction effect between groups and time, independent t-test between groups and contrast for baselines by group were conducted to determine significance.

Point 6: Are the body weights in HFD and HFD+EX, ND and ND+EX significantly different? The authors write this in the text (lines) but do not show or explain properly in figure 2. 

Response 6: It was added in the results description as follows.

‘The ND + EX group showed significantly lower body weight than the ND group from first week (7th weeks in total) after the start of exercise treatment, and The HFD + EX group showed no significant difference from the ND group from the 8th week of exercise treatment (14th weeks in total). Through these results, it was possible to confirm the exercise effect of clear weight loss.’

Point 7: The graph in figure 2 should also report the standard error.

Response 7: The SE was added and Figure 2 was revised as follow.

Point 8: The western blot in figure 3A seems not in agreement with the quantitation of IL-8 in graph 3C. The immunoband in ND+EX seems less intense than HFD (and GAPDH is similar for all the 4 samples).

It would be interesting and helpful to calculate the ratio of IL-8/CXCR2 to understand whether the receptor expression paralleled the increase of IL-8

Response 8: The IL-8 figure3A has been identified and corrected.

Point 9: The discussion on IL-6 seems to me rather long. The main conclusion is that IL-6 does not change and is in agreement with previous data reported in the literature.

 Response 9: It was summarized in the discussion and was revised as follows.

IL-6 is a typical myokine known to regulate multiple physiological functions after exercise training, and activates signal transducer and activator of transcription 3 signaling in human satellite cells after muscle-lengthening contraction [22]. Moreover, IL-6 has important effects on hypertrophic muscle growth and myogenesis in mice [21]. In this study, IL-6 protein expression was appreciably lower in the skeletal muscles of obese rats compared to those fed a normal diet; however, although IL-6 increased slightly in obese rats after exercise training, the effect was not significant. Consistent with this, previous studies reported no significant changes in IL-6 mRNA levels after 11 weeks of exercise training [18] nor on plasma IL-6 levels after 12 weeks of endurance training [23]. Taken together, there appears to be limited evidence to suggest that IL-6 levels in skeletal muscles are influenced by exercise [7]. Patients with chronic obstructive pulmonary disease did not show changes in IL-6 and tumor necrosis factor-β mRNA levels in their skeletal muscles after 10 weeks of exercise [24], while their IL-4 and IL-13 mRNA levels did increase [25]. Moreover, seven weeks of exercise changed the expression patterns of IL-1 receptor antagonist, IL-1, and IL-12 in the skeletal muscles or fatty tissues of rats, but no distinct changes in IL-6 protein expression were observed [11]. However, Knudsen et al.[26] suggested that IL-6 plays important roles regulating substrate utilization in skeletal muscle, basal and exercise-induced adaptations in adipose tissue glucose uptake, and lipolysis during recovery from exercise. To definitively identify the effects of exercise training on muscle IL-6 protein expression, additional studies that consider the detailed contents of the exercise program and the specific characteristics of the subjects will be necessary.

The secretion of IL-6 from skeletal muscle after exercise has important effects on the regulation of endocrine function, playing roles in insulin-stimulated glucose disposal and increased glucose oxidation [27], and stimulating lipolysis and fat oxidation [28,29]. At the molecular level, such effects are mediated by the IL-6-dependent activation of AMP-activated protein kinase [30], insulin receptor substrate-1 [31], and phosphoinositide 3-kinase [32]. Moreover, IL-6 also facilitates the alternative activation of macrophages that have been restrained in tissues by obesity-induced insulin resistance and inflammation [33]. Recently, it has been reported that exercise can reduce tumor size and growth in mice through IL-6-dependent recruitment of natural killer cells [34]. In other words, IL-6 secretion after exercise training can have beneficial effects, including blood sugar regulation, lipolysis, inhibition of tumor growth, and maintenance of muscle mass. However, IL-6 is regulated by carbohydrate availability and acts as a sensor of the metabolic status of muscles [35,36]. Therefore, since it has been reported that IL-6 and IL-8 transcription levels after exercise when muscle collagen content is low [6], additional future studies are needed to uncover the effects of detailed exercise training programs and diet compositions on IL-6 expression in elderly and obese subjects.

  • IL-6 is a typical myokine known to regulate multiple physiological functions after exercise training, and activates signal transducer and activator of transcription 3 signaling in human satellite cells after muscle-lengthening contraction [16]. Moreover, IL-6 has important effects on hypertrophic muscle growth and myogenesis in mice [15]. In this study, IL-6 protein expression was appreciably lower in the skeletal muscles of obese rats compared to those fed a normal diet; however, although IL-6 increased slightly in obese rats after exercise training, the effect was not significant. Consistent with this, previous studies reported no significant changes in IL-6 mRNA levels after 11 weeks of exercise training [14] nor on plasma IL-6 levels after 12 weeks of endurance training [17]. Taken together, there appears to be limited evidence to suggest that IL-6 levels in skeletal muscles are influenced by exercise [7]. However, the secretion of IL-6 from skeletal muscle after exercise has important effects on the regulation of endocrine function, playing roles in insulin-stimulated glucose disposal and increased glucose oxidation [24], and stimulating lipolysis and fat oxidation [25,26]. At the molecular level, such effects are mediated by the IL-6-dependent activation of AMP-activated protein kinase [27], insulin receptor substrate-1 [28], and phosphoinositide 3-kinase [29]. Moreover, IL-6 also facilitates the alternative activation of macrophages that have been restrained in tissues by obesity-induced insulin resistance and inflammation [30]. In other words, IL-6 secretion after exercise training can have beneficial effects, including blood sugar regulation, lipolysis, inhibition of tumor growth, and maintenance of muscle mass. To definitively identify the effects of exercise training on muscle IL-6 protein expression, additional studies that consider the detailed contents of the exercise program and the specific characteristics of the subjects will be necessary.

Point 10: How can the decrease of the VEGF in ND+EX group be explained?

Response 10: Actually, we didn't know the cause clearly, so we added the following sentence.

‘Especially the exact cause of the decrease in the ND + EX group compared to the ND group in relation to VEGF expression could not be confirmed and should be left as a research problem for the next study.’

Minor points :  

Point 11: Figure 1: climb down is not correct

Response 11: We revised to ‘Ladder climbing’

Point 12: Line 97: slope or inclination is more appropriated than incline

Response 12: We could not find your check.

Point 13: Line 112: Clipped left ventricle? I think this sentence is not correct

Response 13: We could not find your check

Point 14: The references present some typos errors.

Response 14: References were revised according to the format as a whole.

Round 2

Reviewer 1 Report

Authors revised several paragraphs and sentences to improve the quality of their manuscript. However, the results with simple evaluation of protein expression in skeletal muscles and body weight changes during the experimental period could not support authors' conclusion. Still, other supportive results or evidences for conclusion are needed.

Author Response

Response to Reviewer 1 Comments

Point 1: Authors revised several paragraphs and sentences to improve the quality of their manuscript. However, the results with simple evaluation of protein expression in skeletal muscles and body weight changes during the experimental period could not support authors' conclusion. Still, other supportive results or evidences for conclusion are needed.

Response 1 : Considering what was pointed out, it was revised to a limited conclusion within the scope of the results obtained in this study as follow.

Abstract :

In conclusion, obesity resulted in decreased myokines in the skeletal muscles of aging rats, but exercise training increased myokines secretion, suggesting its promise in reducing inflammation and enhancing immune function during obesity.

-> In conclusion, high fat diet-induced obesity resulted in decreased myokines in the skeletal muscles, but combined exercise training of aerobic and resistance exercise increased myokines secretion in skeletal muscle of obese rats, and is thought to help reduce inflammation.

  1. Conclusions

In conclusion, high fat diet-induced obesity suppressed the expression of IL-7, IL-8, IL-6, CXCR2, and VEGF in the skeletal muscles of aging obese rats; however, a 12-week program combining aerobic and resistance exercise increased the secretion of IL-7, IL-8, CRCX2, and VEGF, indicating that exercise is a useful means of reducing inflammation.

-> In conclusion, high fat diet-induced obesity suppressed the protein expression of IL-7, IL-8, IL-6, CXCR2, and VEGF in skeletal muscle. However, combined exercise training of aerobic and resistance exercise increased myokines secretion in skeletal muscle of obese rats, and is thought to help reduce inflammation.

Reviewer 2 Report

The Authors responded and modified the manuscript according to most of the comments raised. However, the manuscript needs text editing

Author Response

Response to Reviewer 2 Comments

Point 1: The Authors responded and modified the manuscript according to most of the comments raised. However, the manuscript needs text editing.

Response 1 : We've edited the whole new one, and we'll edit the missing part additionally. The text was revised through text editing (Line spacing, paragraph type, and misspelling

etc), and the font was unified to Paltino Linotype.
